# An Intervention of 12 Weeks of Nordic Walking and Recreational Walking to Improve Cardiorespiratory Capacity and Fitness in Older Adult Women

**DOI:** 10.3390/jcm11102900

**Published:** 2022-05-20

**Authors:** Nebojsa Cokorilo, Pedro Jesús Ruiz-Montero, Francisco Tomás González-Fernández, Ricardo Martín-Moya

**Affiliations:** 1Faculty of Sport, University UNION Nikola Tesla, 11158 Belgrade, Serbia; cokorilon@gmail.com; 2Physical Education and Sport Department, Faculty of Education and Sport Sciences, Campus of Melilla, University of Granada, 52005 Melilla, Spain; ftgonzalez@ugr.es (F.T.G.-F.); rmartinm@ugr.es (R.M.-M.)

**Keywords:** ageing, physical fitness, walkers, well-being, physical exercise programme

## Abstract

(1) Background: The main aim of this study was to examine the effect of an intervention of 12 weeks in three groups on anthropometric measurement and heart rate (HR) variables, fitness index, and maximal oxygen consumption (VO_2_max) in older women. (2) Methods: In total, 166 Serbian adult women, aged 50 to 69 years old, participated in this study, comprising a control group (60 participants, μ_age_ = 57.8 + 6.6), Nordic-walking (NW) group (53 participants, μ_age_ = 57.5 + 6.8), and recreational-walking (RW) group (53 participants, μ_age_ = 57.8 + 6.6) in a physical fitness programme for 12 weeks. (3) Results: Anthropometric measurement variables were measured using a stadiometer and an electronic scale. The data showed differences in walking heart rate (bt/min) (*p* < 0.001; η^2^ = 0.088) between control, NW, and RW groups in the pretest analysis. Moreover, there were significant differences in walking heart rate (bt/min) (η^2^ = 0.155), heart rate at the end of the test (bt/min) (η^2^ = 0.093), total time of fitness index test (min) (η^2^ = 0.097), fitness index (η^2^ = 0.130), and VO_2_max (η^2^ = 0.111) (all, *p* < 0.001) between control, NW, and RW groups in the posttest analysis. (4) Conclusions: NW group training resulted in slightly greater benefits than RW group training. The present study demonstrated that both groups could act as modalities to improve the functionality and quality of life of people during the ageing process, reflected mainly in HR variables; UKK test measurements, and VO_2_max. It also contributes to the extant research on older women during exercise and opens interesting avenues for future research.

## 1. Introduction

In recent decades, growing evidence from various experimental approaches has shown that the ageing process with a sedentary lifestyle (e.g., sitting and engaging with television and other electronic devices) leads to a greater likelihood of suffering health problems [1]. The majority of this research shows that more sedentary time is linked with increased risk for cardiovascular disease [2] and adverse metabolic effects such as obesity and insulin resistance in women, among other effects [3]. In fact, global health organisations such as the World Health Organisation (WHO) and the American College of Sports Medicine (ACSM) report that sedentary life in adult and older adult women is apparently related to inactivity behaviour during occupational and domestic activities and little participation in physical exercise (PE) during leisure and free-time activities [4].

Previous studies indicate that ageing and a low level of PE are directly related to cardiovascular and pulmonary changes that lead to a reduction in functional capacities [5,6,7]. In this sense, different physiologic mechanisms that affect health are caused by sedentary time. For example, sedentary time decreases the metabolic activity of muscle and decreases energy utilisation, causing insulin resistance and metabolic disorders [8]. Therefore, chronic PE could help to combat a sedentary lifestyle. In fact, numerous studies have shown the beneficial effects of PE on overall health, specifically in physical fitness and anthropometric measurement [9,10,11]. Recent systematic reviews and meta-analyses have concluded that physical exercise helps to reduce the risk of various conditions associated with ageing, such as frailty [12], cognitive decline [13], low muscle power, or poor functional capacity [14]. There is also evidence that exercise intervention programmes can prevent falls in older people with mild comorbidities [15]. However, the literature shows that, in addition to preventing functional capacities, PE also prevents mental diseases and supports health benefits across the older adult lifespan [16]. In addition, it is well-documented that changes in body composition and anthropometric measurement are strongly related to ageing; the main effects are lean body mass and fat mass [17,18]. In this respect, variations in body composition and anthropometric measurement in older people have also been associated with loss of muscle mass and, therefore, with muscle strength, physical capacity, and quality of life and well-being [19,20].

Other important physiologic mechanisms that influence ageing are the heart rate (HR) and maximal oxygen consumption (VO_2_max). One study showed that age-related declines in VO_2_max were approximately 0.35, 0.44, and 0.62 mL/kg/min per year for sedentary, physically fit, and physically trained females, respectively [21]. In this sense, an intervention of 16 weeks of combined aerobic exercise training and resistance exercise training produced a significant improvement in muscular strength, cardiovascular fitness, and functional tasks in older women [22]. Overall, the majority of effects of ageing on the heart can be decreased by chronic exercise; therefore, PE can help people maintain cardiovascular fitness as well as muscular fitness as they age [23].

Healthy ageing is the key to maintaining an adequate level of physical performance, which is needed to be able to successfully perform everyday activities [24]. Currently, there is a large number of tests whose objective is to assess the physical condition of different population groups. From the point of view of health care and biomedical fields, tests have the main objective of prescribing a PE programme according to individual characteristics. In this sense, obtaining correct values could help to assess the initial fitness level of each subject. Urho Kaleva Kekkonen (UKK) test has a very simple application, reliability, health-related validity, and physical-activity-related safety and feasibility [25], especially for adults aged 18–69. In addition, the UKK test has a large correlation with VO_2_max [25]. Due to having all these properties, the UKK test may show the variability of physical performance over time and provide evidence for maintaining good physical fitness and promoting healthy ageing in different age groups [26].

As mentioned above, a low level of PE among older adults, caused by a sedentary lifestyle, leads to reduced functional capacities and, therefore, damage to health over time. To overcome this issue, the literature suggests that individuals practice regular PE of low-to-moderate intensity, such as recreational walking (RW) or Nordic walking (NW), which may encourage older people to practice regular PE [27,28]. RW may be a simple and efficient means of PE for older people due to the low risk of injury, and irrespective of individual level [29], it is an effective way to improve the level of PE, requires no equipment, and can be performed almost anywhere at any time. Thus, RW involves moderate intensity that can provoke positive changes to health and minimise the risk of premature death [30]. With regard to NW, a systematic review reveals its enormous positive benefit on health [31]. NW has grown in popularity recently and is considered a good PE for older people due to its safety and low cost. Originally derived from RW, NW is highly popular and accepted by the population [32]. The literature has shown that NW reduces the load of the lumbar spine and lower limb joints and increases energy expenditure in comparison with RW [33]. Finally, the beneficial effects of NW on different health parameters such as resting HR, blood pressure, exercise capacity, and VO_2_max have been established in different populations [34,35]. In addition, recent studies have shown the effectiveness of programmes of regular PE based on NW [36]. Taking into account the current literature, there is strong scientific evidence that regular physical activity has extensive health benefits for adults aged 65 and above, but there is little evidence to suggest which are the most suitable practices if we talk about Nordic-walking or recreational-walking modalities [34], and how they can serve to improve the physical condition and quality of life of older adults. In fact, older adults perceive them as easy and enjoyable types of exercise, and they provide effective ways to promote an active lifestyle for improved health. Thus, the aim of the present study was to examine the effect of a 12-week intervention in three groups (control, RW, and NW groups) on anthropometric measurement variables (body mass and body mass index (BMI)), HR variables (walking HR and HR at the end of the test), UKK test measurement (total time and fitness index), and VO_2_max in older women.

## 2. Materials and Methods

A total of 166 older women (50–69 years; μ_age_ = 57.6 ± 5.6) from the north of Serbia (city of Novi Sad) were recruited through social groups from Novi Sad University (Serbia), leaflets, local newspapers, and social media. Specifically, there were 60 participants in the control group (μ_age_ = 57.75 ± 3.51), 53 participants in the RW group (μ_age_ = 57.85 ± 6.64) and 53 in the NW group (μ_age_ = 57.53 ± 6.85) (see Figure 1). As initial contact with potential recruits yielded only seven men, researchers decided to recruit only female participants in the study to avoid problems of statistical power with a low male sample.

Based on a statistical power of 80% (z), a type 1 margin of error or alpha of 0.05, a response distribution of 50% (r), and a sample population of older women (50–69 years) (N_recommended_ = 164) in the city, the sample size of the present study was in the recommended range. The following formulas were used [37]:x = Z(c/100) 2r (100 − r)
n = N x/((N − 1) E2 + x)
E = Sqrt [(N − n) x/n(N − 1)]

Anthropometric measurement variables (body mass and BMI), HR variables (walking HR and HR at the end of the test), UKK test measurements (total time and fitness index), and VO_2_max were gathered.

All participants were provided with a written document that specified the research objective. They were recruited via telephone or direct contact from an association of women that used to gather together in the Faculty of Sport Science of Novi Sad to engage in several leisure activities (not involving physical activity or sport). We informed all participants not to modify their daily behavioural patterns and not to engage in other extra physical exercises, to avoid conflation of results. A sampling strategy was determined for convenience.

The inclusion criteria for all women who participated in this study were (1) 50–69 years of age, (2) no severe somatic or psychiatric disorders, (3) ability to complete the walking test of two kilometres without assistance, and (4) oral and written communications ability. Exclusion criteria for this study were (1) not to be diagnosed with an acute or terminal illness; (2) to have suffered a major cardiovascular event (i.e., myocardial infarction, angina, or stroke) in the past 6 months; (3) presence of neuromuscular disease or drugs affecting neuromuscular function, (4) unwillingness to complete the study requirements, and (5) presence of neuromuscular disease or drugs affecting neuromuscular function.

This study was conducted in accordance with the ethical principles of the Declaration of Helsinki, and it was approved by the Ethical Committee (no. 46-06-03/2020-1), with the purpose of ensuring a responsible investigation. Moreover, this study was registered as a controlled and non-randomised design in ISRCT (ISRCTN44310625).

### 2.1. Measures

#### 2.1.1. Anthropometric Measurement Variables

Age and gender were determined by two ad hoc questions. Body height was measured to ±0.1 mm using a stadiometer (SECA 213, Hamburg, Germany), and body mass (kg) was measured with an electronic scale (SECA 799, Hamburg, Germany), with participants wearing light indoor clothing and no shoes. BMI was calculated as weight (kg)/height^2^ (m). Prior to measurement, participants were asked to have fasted for 4 h, to not have consumed alcohol for 8 h, and to not have performed physical exercise for the previous 8 h. All tests were repeated in the same space and at the same time after 12 weeks under the same humidity condition (30–40%), as recommended by Morente-Oria et al. [24]. BMI was calculated as kg/m^2^.

#### 2.1.2. Cardiorespiratory Fitness

To measure the cardiorespiratory capacity, achieved by brisk walking and prediction of VO_2_max, the UKK walking test for adults aged 18–69 [25], developed by the Urho Kaleva Kekkonen Institute for Health Promotion Research (the UKK Institute), was used. The UKK walking test involves walking two kilometres on a flat surface at as brisk a pace as possible. The results of this test indicate a fitness index, taking into consideration the age, gender, body mass, duration of the walk, and HR at the end of the test. Furthermore, we measured the walking HR average over the two kilometres. The fitness index by cardiorespiratory capacity fitness index uses five levels, where the lowest level is <70 (considerably below average) and the highest is >130 (considerably above average). The equation to calculate UKK fitness index by the cardiorespiratory capacity of women was as follows [38]:

1º step: (walking time-min * 8.5) + (walking time-seconds * 0.14) + (HR at the end of the test-beats/minutes * 0.32) + (BMI-kg/m^2^ * 1.1) = X; 2º step: X − (age * 0.4) = Y; 3º step: 304 − Y = UKK fitness index.

To calculate the VO_2_max, an equation predicting maximal aerobic power on the basis of the results obtained in the UKK walking test was used for women in this study as follows [38]:

116.2 − 2.98 * time(minutes/seconds) − 0.11 * HR (at the end of the test) − 0.14 * age − 0.30 * BMI (kg/m^2^).

#### 2.1.3. Heart Rate (HR)

The HR at the end of the UKK walking test and the walking HR during the UKK walking test were measured by a pulse watch and chest belt (Polar FT2, Kempele, Finland). The measured values were obtained immediately after finishing the pretest and posttest intervention evaluations of the UKK walking test.

#### 2.1.4. Patient and Public Involvement

Data were collected twice: a test performed during the month of August 2021 (pretest) and after 12 weeks (posttest), performed in October 2021.

The participants visited the measuring area at the University of Novi Sad (Serbia), and their anthropometric measurements were measured 48 h before starting the interventional programme and 48 h after the programme. They were evaluated in the morning, and environmental conditions were controlled.

Both training programmes (RW and NW) were performed three times a week for 12 weeks (3 months) in the morning, with a duration from 35 to 45 min, in line with the requirements laid out by the ACSM [4]. During this intervention, participants took part in no other physical activity.

The participants from the experimental NW group used specific telescope aluminium poles (100–135 cm) for 3 months. Participants from the experimental RW group used no poles.

The programme was carried out on a trim track (park “Sremska Kamenica”, Novi Sad, Serbia), which offered good conditions for this activity. The training programme was supervised by physical exercise specialists and adapted individually for each of the participants, depending on participant age and in compliance with sports training principles. In addition, each specialist controlled a different studied variable (anthropometric measurement, HR, UKK test measurements, and VO_2_max variables. Thus, any bias related to assessors and researchers was minimum. Participants’ individual physical limitations were considered in controlling for the range and intensity of exercise.

Prior to each activity, participants were made aware of the necessary HR during the training, and the programme was conceptualised in such a way that participants were always in the aerobic zone of performance. During the walking exercise, HR was monitored by a pulse meter and was used to determine load intensity. Both programmes were divided into three parts, different by volume (frequency of sessions per week and length) and intensity (percentage of maximum heart rate (%HRmax) according to the age and physical activity level of participants). Participants performed three weekly training sessions of 35 min of continuous aerobic work during the first 4 weeks (first month of the training programme), and the intensity was 60–65% of the total. From the fifth to eighth training week (second month of the training programme), the duration of training sessions was 40 min of continuous aerobic work three times per week at 65–70% %HRmax. Finally, in the last 4 weeks (third month), the sessions were 45 min in length three times per week, with an %HRmax of 75–80%. The differences in burdening (training intensity) according to age were calculated by using the percentage of maximum heart rate and optimal intensity of burdening within the limits of 50–90% [39] (Table 1).

For the RW and NW groups, participants had to do a 10 min warm-up in a dedicated area of the trim track. Specifically, the participants performed movement exercises with different specific gestures such as lateral steps, knee elevation, tiptoe walk, and fast arms movement. After the warm-up, the main block of the session was followed by 20 min (weeks 1–4), 25 min (weeks 5–8), and 30 min (weeks 9–12) of continuous RW or NW depending on the experimental group. This was followed by a 5 min cool-down period, featuring mainly stretching exercises. Participants of the control group did not undergo supervised training and were asked not to modify daily activities. However, as physical exercise and active life are beneficial to health, participants in the control group attended three workshops (one per month) addressing the benefits of being physically active. The attendance at these workshops aimed to maintain participation in the control group and their commitment to completing the entire 12 weeks of intervention.

Finally, participants received a report of the results and conclusions obtained in the study after the elaboration and discussion of the data, thanking them for their participation.

#### 2.1.5. Statistical Analysis

The Kolmogorov–Smirnov test with mean values was used to test the normality of distribution. The mean (μ) and standard deviation (DT) of the anthropometric measurement variables (body mass and BMI), HR variables (walking HR and HR at the end of the test), UKK test measurements (total time and fitness index), and VO_2_max of the participants were determined. In addition, test reliability was determined by calculating the coefficient of variation of VO_2_max between pretest and posttest in the RW and NW groups.

A mixed-design ANOVA for participants was used between groups (control, Nordic walking, and recreational walking) and moment intervention (pretest–posttest). To establish the differences in outcomes, we used a one-way analysis of variance (ANOVA) in the pre- and posttest. Pairwise comparisons were performed with Bonferroni’s adjustment. Effect size is indicated with partial eta squared (η^2^) for Fs [40]. The Greenhouse–Geisser correction was applied when sphericity was violated [41]. In such a case, corrected probability values were reported. Differences in participant groups were found in the five categories of UKK fitness index. Regarding the pretest and posttest differences among all groups, a repeated-measures ANOVA test was used for two dependent samples. This process was carried out with regard to anthropometric measurement variables, HR variables, UKK test measurements, and VO_2_max in all groups.

All statistical analyses were performed using the Statistical Package for Social Science (IBM SPSS Statistics for Windows 21.0. Armonk, NY, USA).

## 3. Results

### 3.1. Comparison of the Participant Groups (Control, RW, and NW) Relative to Anthropometric Measurement Variables, HR Variables, UKK Test Measurements, and VO_2_max

Table 2 shows only one significant difference among the three participant groups in the pretest. The walking HR was lower in the RW group than that in the control and NW groups (79.17 ± 9.45 bt/min vs. 82.13 ± 9.36 bt/min and 87.12 ± 12.36 bt/min, respectively, *p* < 0.001).

However, Table 3 reveals several significant differences among the control, NW, and RW groups, such as the walking HR (82.35 ± 9.09 bt/min vs. 79.45 ± 9.36 bt/min and 73.54 ± 7.32 bt/min, respectively, *p* < 0.001) and HR at the end of the test (114.13 ± 11.56 bt/min vs. 105.09 ± 11.61 bt/min and 106.51 ± 14.81 bt/min, respectively, *p* < 0.001). The results of measurements of the UKK test in terms of the UKK total time spent to walk two kilometres and the fitness index were significantly worse in the control group than those for the NW and RW groups, as detailed below. Participants in the RW group spent the shortest time to walk two kilometres (RW group = 22.06 ± 1.91 min; NW group = 22.11 ± 1.76 min; control group = 23.32 ± 1.84 min; *p* < 0.001) and obtained the best UKK fitness index (RW group = 88.92 ± 19.42; NW group = 84.72 ± 15.94; control group = 72.95 ± 18.13; *p* < 0.001). Finally, the RW group showed higher VO_2_max than the NW and control groups (24.51 ± 6.41 mL·min^−1^·kg^−1^ vs. 23.58 ± 6.05 mL·min^−1^·kg^−1^ and 19.59 ± 6.14 mL·min^−1^·kg^−1^, respectively, *p* < 0.001). According to the coefficient of variation of VO_2_max, it is possible to observe an increase of 23.59% in the difference between pretest and posttest results in RW and 30.20% in NW.

### 3.2. Differences in Body Composition Variables, HR Variables, UKK Test Measuring, and VO_2_max between Pretest and Posttest by Participant Group (Control, RW, and NW)

A mixed-design ANOVA with mean data of body mass (F = 410.13, *p* < 0.001, η^2^ <= 0.72) and BMI (F = 412.63, *p* < 0.001, η^2^ = 0.71) revealed a significant main effect of participant groups (control, RW and NW). The interaction between group and pretest–posttest intervention showed a significant effect in body mass (F = 113.33, *p* < 0.001, η^2^ = 0.58) and BMI (F = 111.73, *p* < 0.001, η^2^ = 0.57). Moreover, a repeated-measures ANOVA test between pretest and posttest in the RW and NW groups demonstrated differences in body mass (RW group, *t* = 14.49, *p* < 0.001; NW group, *t* = 15.09, *p* < 0.001), BMI (RW group, *t* = 14.37, *p* < 0.001; NW group, *t* = 15.16, *p* < 0.001), and walking HR (RW group, *t* = −14.36, *p* < 0.001; NW group, *t* = −9.41, *p* < 0.001). This is shown in Figure 2.

The mean data of walking HR (F = 62.94, *p* < 0.001, η^2^ = 0.28) and the HR at the end of the UKK walking test (F = 82.59, *p* < 0.001, η^2^ = 0.31) determined by a mixed-design ANOVA revealed a significant main effect of participant groups (control, RW and NW). The interaction between group and pretest–posttest intervention showed a significant effect in walking HR (F = 19.25, *p* < 0.001, η^2^ = 0.19) and the HR at the end of the test (F = 21.77, *p* < 0.001, η^2^ = 0.21). The HR at the end of the test showed differences between the pretest and posttest (RW group, *t* = 8.99, *p* < 0.001; NW group, *t* = 4.45, *p* < 0.001), while the walking HR during the test also showed significant differences (RW group, *t* = 5.90, *p* < 0.001; NW group, *t* = 5.60, *p* < 0.001) (Figure 3).

The mean data of UKK total time (F = 183.01, *p* < 0.001, η^2^ <= 0.53) and UKK fitness index (F = 132.81, *p* < 0.001, η^2^ <= 0.45) determined by a mixed-design ANOVA revealed a significant main effect of participant groups (control, RW and NW). The interaction between group and pretest–posttest intervention showed a significant effect in UKK total time (F = 47.55, *p* < 0.001, η^2^ <= 0.36) and UKK fitness index (F = 33.55, *p* < 0.001, η^2^ < = 0.29). The significant differences between before and after 12 weeks of intervention in terms of UKK test measurements are shown in Figure 4—namely, UKK total time (RW group, *t* = 13.03, *p* < 0.001; NW group, *t* = 8.29, *p* < 0.001) and UKK fitness index (RW group, *t* = −6.79, *p* < 0.001; NW group, *t* = −6.41, *p* < 0.001).

Finally, the results of the mixed-design ANOVA for VO_2_max (F = 230.89, *p* < 0.001, η^2^ <= 0.58) showed the significant main effect of participant groups (control, RW, and NW). The interaction between groups and pretest–posttest intervention also showed a significant effect in VO_2_max (F = 60.91, *p* < 0.001, η^2^ <= 0.43). Figure 5 shows significant differences in VO_2_max between pretest and posttest in the RW group (*t* = −14.36, *p* < 0.001) and the NW group (*t* = −9.40, *p* < 0.001). The control group showed no significant differences between pretest and posttest interventions in any variable studied.

## 4. Discussion

The objective of the present study was to examine the effects of a 12-week intervention in three groups (control, RW, and NW groups) on anthropometric measurement variables (body mass and BMI), HR variables (walking HR and HR at the end of the test), UKK test measurements (total time and fitness index), and VO_2_max in older adult women. Given the specific nature of exercise adaptations and the need to maintain muscle mass, muscle strength, and flexibility throughout life, and even more in the ageing process, a comprehensive training programme consisting of resistance, aerobic, and flexibility exercises is recommended [4]. NW can improve muscle strength and flexibility, as well as aerobic endurance capability [35]. Likewise, RW, together with a muscular strength programme, can produce the same improvements as the NW discipline, because the disciplines do not differ considerably [29]. The main finding of this study showed that NW group training resulted in slightly greater benefits than RW group training; nevertheless, both were shown to be valid modalities to improve the functionality and quality of life of people during the ageing process.

In regard to the variable of walking HR, we found that the lowest mean was presented by the RW group, followed by NW. In accordance with a recent systematic review and meta-analysis carried out by Bullo et al. [34], these data contradict the majority of studies. These results could be due to the fact that it has been shown that healthy older adults can maintain their gait speed with or without poles, thus matching the physical demands required [36]. The same results were found in a study by Takeshima et al. [33]. Both disciplines showed improvements after the training programme. These data are in line with those found for participants of the same age range in the literature [28] or for those registered in control groups of physical training programmes [37], which confirms favourable results in the experimental groups of this study. We observed that, after the intervention, RW produced better values than NW in terms of walking HR and HR at the end of the test; NW values were also significant and showed improvements very similar to those of the RW modality. These data are in accordance with those found by Gomeñuka et al. [35], in which the NW discipline did not result in greater benefits than RW related to participants’ quality of life.

As expected, the improvements in the average walking HR were greater in the NW group than in the RW group at the end of the program. These data could be explained by the fact that NW increases the activity (muscular) of the upper part of the body during practice and, with it, the energy expenditure with higher oxygen consumption [38]. If we compare our results with the findings of Pellegrini et al. [28], it can be deduced that NW group participants of the present study showed lower bt/min during the 2 km test. However, in a study by Sugiyama et al. [42], participants were evaluated for 5 km, whereas we studied only 2 km. Therefore, increases in both strength and aerobic capacity contribute to the reduction in submaximal HR when NW and RW are performed at the same walking speed [39]. In the present study, UKK total time was similar for both disciplines. Regarding the variable HR at the end of the test, the NW group presented the best average. In a similar way, we found comparable results for RW, elucidating how both modalities generated improvements in the reduction in mean beats per minute at the end of the programme [35]. Similar improvements were shown for the rest of the variables described above and below (HR, fitness index, and VO_2_max).

NW and RW are two aerobic activities capable of improving anthropometric measurement through the ageing process, and a 12-week programme using these two modalities, such as that used in the present study, results in significant changes in terms of body mass loss and BMI [41,43]. Although both exercise disciplines showed comparable results, we found that RW produced slightly better results than NW in body mass and BMI after 12 weeks of intervention. According to a study by Figard-Fabre et al. [44], even with a correct NW technique, the difference in energy expenditure is not greater than that in RW. This minimal difference in energy expenditure could explain how RW obtained slightly better results in the current study, underlining that NW has also shown improvements in this regard. These results are not in line with the fact that energy expenditure is higher in the NW pattern, and this would mean a greater body mass loss [28,45].

Given the nature of the NW modality, gait speed is greater with the use of poles versus walking without any help [46]. Although the time to complete the activity was similar, the RW group presented a slightly shorter time. This is consonant with data found by Gomeñuka et al. [35] and contrary to what is found in other studies [31,33,46]. This could be explained by the fact that if the poles do not push the ground in the correct way for the propulsion of the movement, the dynamic stability is impaired; thus, the movement is not as efficient and fast, requiring more time to complete a given distance [47].

In the final measurement, UKK fitness level and VO_2_max showed slightly higher values in the RW group compared with the NW group, which points to a greater improvement in functional capacities in the former group, compared with the control group. This increase was manifested as an improved UKK fitness level, increased general cardiorespiratory capacity, and higher percentages of VO_2_max, which were considered the best indicators of lung capacity and air intake. A study by Pellegrini et al. [46] reported NW and RW as acceptable forms of exercise for the ageing population regardless of their VO_2_max. The results found in this study would contradict those shown in the literature, since, as previously described, NW generally showed higher energy expenditure and, with it, the expectation that UKK fitness level and VO_2_max values should also be higher in this group [46]. These data could be explained by the fact that if participants’ NW technique was incorrect, resulting in a weak NW style instead of the suggested technique, this could decrease the effectiveness of the training programme [28]. Hansen and Smith [48] found that when participants are taught to perform a particularly energetic NW technique, characterised by long strides and pole pushes, oxygen consumption exceeds that measured during walking, by up to 67%. In contrast, in line with the findings of the present study, NW and RW showed similar results, which can be explained by the fact that both modalities share muscle synergies that mainly involve the activation of the muscles of the lower extremities and the trunk. Therefore, walking with the use of poles does not produce considerable changes in the muscular demands of the lower body [31].

If each modality and variable is analysed separately, NW produced greater improvements with respect to RW. In a study carried out by Mikalački et al. [49], in which they evaluated the effects of a 12-week NW programme on functional capacities using the UKK test, results showed improvements in UKK fitness values, which is consistent with our findings. Other studies on NW have shown that VO_2_max increases at the end of the intervention [34,50], which is also consistent with our study, showing significant results for NW training in the variables mentioned above.

The outcomes of this study support NW as a secure, viable, and helpful form of physical training that can improve several components of fitness in older adults. RW also has a multitude of other documented benefits, and it is often used as a primary and secondary preventive alternative to PE [4]. Both RW and NW can be performed by individuals or in groups, which may provide social well-being benefits. Additionally, although NW is often performed outdoors, it can also be practised indoors with the use of rubber-tipped poles, turning it into an activity that can be accomplished all year round, even during adverse weather conditions [33]. NW appears to be more effective in providing more cardiorespiratory fitness benefits, compared with RW. Consequently, NW may be more effective than RW in improving and maintaining the overall health and function of adults in the ageing process [51]. Therefore, it is important to know the benefits of both modalities, and it would be convenient to carry out future research in different segments of the older adult population and during longer intervention periods.

In terms of limitations, the present study shows its potential due to the lack of related studies that have compared NW and RW in this population group. On the one hand, the intervention time of 12 weeks should be extended in future research to obtain more significant results, especially in physiological variables and time spent covering the distance of the UKK test. On the other hand, far from being a constraint, the Nordic-walking technique of participants was controlled during the study and supervised by physical exercise specialists. In fact, participants’ individual physical limitations were considered in controlling for the range and intensity of the exercise.

Another variable that could have completed the study would have been to evaluate muscular grip strength and, thus, be able to explain the possible adaptations in terms of upper body strength and physiological changes. Regarding the different groups and the modalities used, it is worth highlighting the need to expand the given consideration to the technique when using poles in NW. Additionally, in the context of control, it would be advisable to control the dietary intake of study participants; if the decrease in body mass is to be taken into account, diet is one of the variables most affected. Likewise, evaluating the improvements in the anthropometric measurement of the participants should include changes in the different variables through electrical bioimpedance.

## 5. Conclusions

The novelty of this study lies in its comparative results between NW and RW interventions for older adults and, with it, guiding healthy alternatives to improve the quality of life and health in the ageing process. The main findings revealed that, after 12 weeks, NW training resulted in slightly greater benefits than RW training; nevertheless, both were shown to be valid modalities to improve the functionality and quality of life of people during the ageing process.

Moreover, our study suggested that the NW technique is a beneficial training method for improving the physical condition of healthy older adults. The practice of the NW and RW modalities represents an optimal continuity of kinesiological activity, particularly in older adults, to maintain and improve their functional capabilities. The present study showed that 12 weeks of Nordic walking and recreational walking improved HR variables, UKK test measurements, and VO_2_max. In fact, this research highlighted the clinical importance of chronic exercise and its considerably positive benefits on health and quality of everyday life. However, caution is required concerning participants’ extensive use without the presence of specialists to control the individual training in older people.

## Figures and Tables

**Figure 1 jcm-11-02900-f001:**
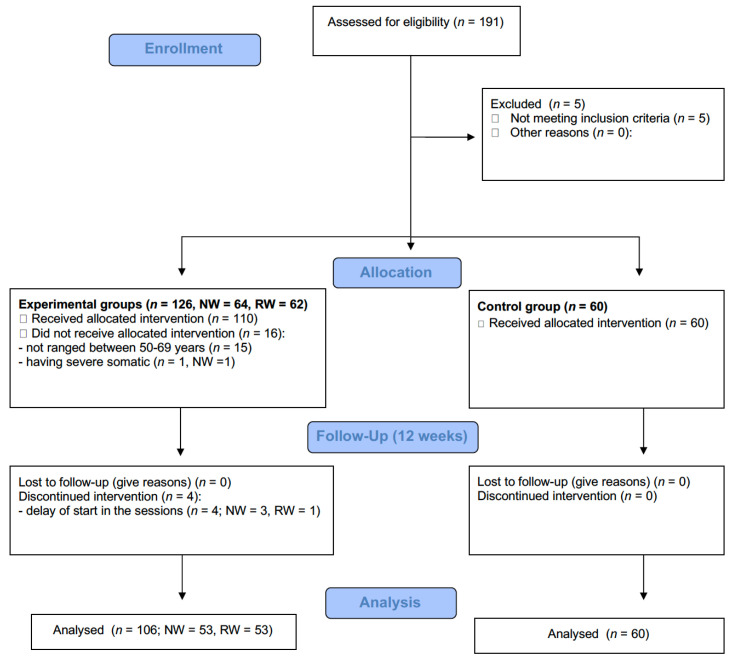
Participant’s flow diagram based on CONSORT reporting guidelines.

**Figure 2 jcm-11-02900-f002:**
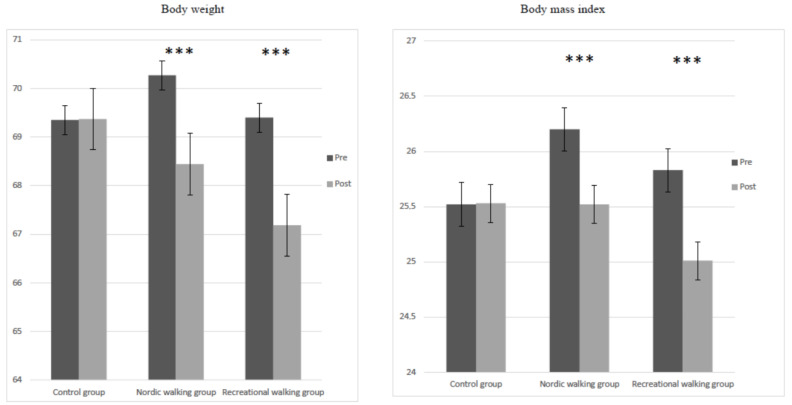
Differences in body weight and body mass index at pretest and posttest interventions in the three studied groups. *** *p* < 0.001.

**Figure 3 jcm-11-02900-f003:**
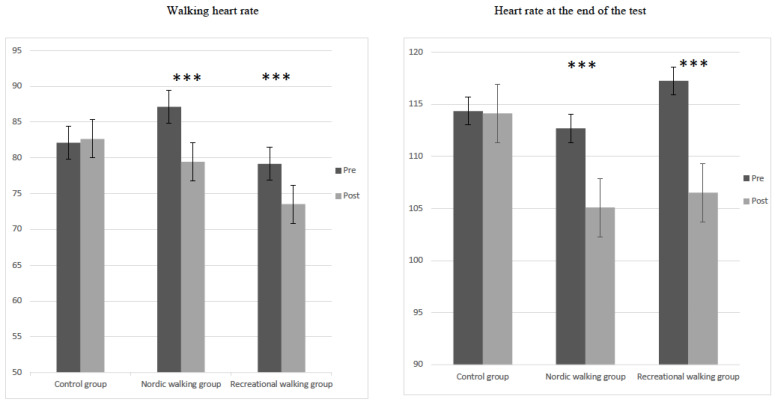
Differences in walking heart rate and heart rate at the end of the test in pretest and posttest interventions in the three studied groups. *** *p* < 0.001.

**Figure 4 jcm-11-02900-f004:**
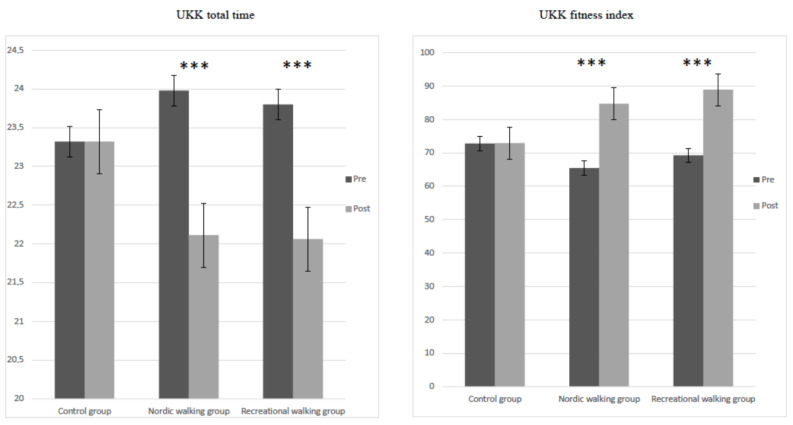
Differences in UKK total time and UKK fitness index in pretest and posttest interventions in the three studied groups. *** *p* < 0.001.

**Figure 5 jcm-11-02900-f005:**
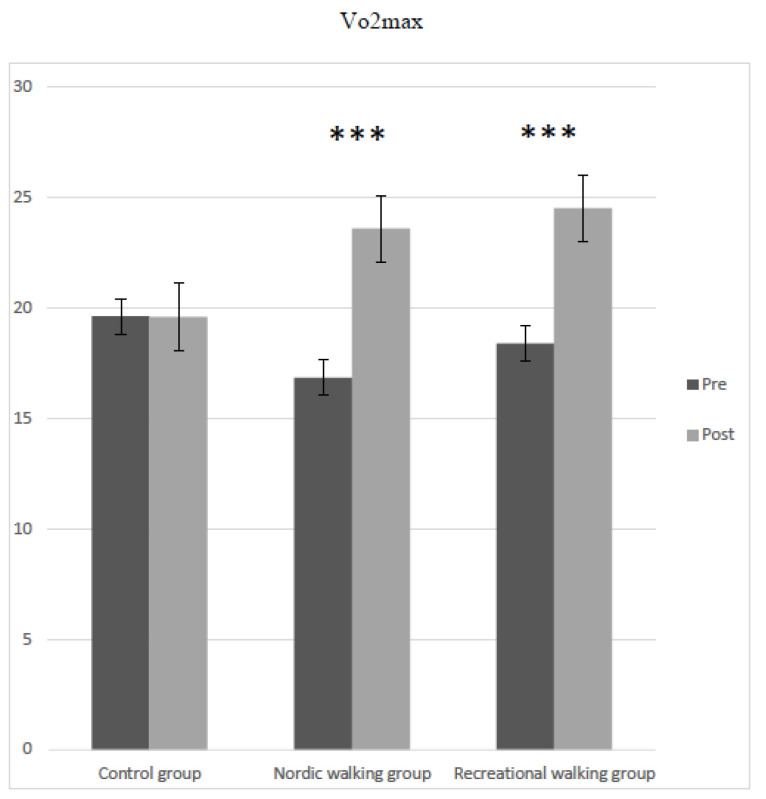
Differences in VO_2_max in pretest and posttest interventions in the three studied groups. *** *p* < 0.001.

**Table 1 jcm-11-02900-t001:** Timeline of the study.

	2021													
	Pretest Intervention	Intervention												Posttest Intervention
Months	July	August				September				October				December
Week		1	2	3	4	5	6	7	8	9	10	11	12	
Control group	Pretest	35 min of continuous aerobic (60–65% %HRmax)												Posttest
Nordic-walking group	Pretest	40 min of continuous aerobic (65–70% %HRmax)												Posttest
Recreational-walking group	Pretest	45 min of continuous aerobic (75–80% %HRmax)												Posttest

**Table 2 jcm-11-02900-t002:** ANOVA for anthropometric measurement variables, heart rate, UKK walking test time and fitness index, and VO2max for pretest, with mean values (SD).

	Pretest					
Variables Involved in UKK Walking Test	Control Group (*n* = 60)	Nordic-Walking Group (*n* = 53)	Recreational-Walking Group (*n* = 53)	F	*p*	η^2^
Body mass (kg)	69.35 ± 9.26	70.27 ± 9.61	69.40 ± 8.38	0.177	0.838	0.002
BMI (kg/m^2^)	25.52 ± 3.58	26.20 ± 3.88	25.83 ± 4.13	0.436	0.648	0.005
Walking heart rate (bt/min)	82.13 ± 9.36 ^b^	87.12 ± 12.36 ^ab^	79.17 ± 9.45 ^a^	7.847	0.001	0.088
Heart rate at the end of the test (bt/min)	114.35 ± 12.48	112.68 ± 15.75	117.26 ± 15.07	1.374	0.256	0.017
UKK total time (min)	23.32 ± 1.65	23.98 ± 1.48	23.80 ± 2.12	2.123	0.123	0.025
UKK fitness index	72.75 ± 15.76	65.42 ± 20.61	69.28 ± 19.01	2.211	0.113	0.026
VO_2_max (mL·min^−1^·kg^−1^)	19.61 ± 5.52	16.85 ± 5.42	18.39 ± 7.31	2.839	0.061	0.034

Note. UKK: Urho Kaleva Kekkonen; BMI: body mass index; VO_2_max: maximal oxygen consumption; m: metre, kg: kilogram; bt: beat; min: minute; mL: millilitre; η^2^: effect size by eta square; SD: standard deviation; df: degrees of freedom (2, 163 for pretest); ^a,b^ superscripts with the same letter show a degree of significant difference between both groups (*p* < 0.05). Pairwise comparisons were performed with Bonferroni’s adjustment.

**Table 3 jcm-11-02900-t003:** ANOVA posttest results in terms of anthropometric measurement variables, heart rate, UKK walking test time and fitness index, and VO_2_max, with mean values (SD).

	Posttest					
Variables Involved in UKK Walking Test	Control Group (*n* = 60)	Nordic-Walking Group (*n* = 53)	Recreational-Walking Group (*n* = 53)	F	*p*	η^2^
Body mass (kg)	69.37 ± 9.25	68.44 ± 9.64	67.18 ± 8.25	0.823	0.441	0.010
BMI (kg/m^2^)	25.53 ± 3.55	25.52 ± 3.85	25.01 ± 4.05	0.332	0.718	0.004
Walking heart rate (bt/min)	82.35 ± 9.09 ^b,c^	79.45 ± 9.36 ^a,b^	73.54 ± 7.32 ^a,c^	14.919	0.001	0.155
Heart rate at the end of the test (bt/min)	114.13 ± 11.56 ^a,b^	105.09 ± 11.61 ^a^	106.51 ± 14.81 ^b^	8.402	0.001	0.093
UKK total time (min)	23.32 ± 1.84 ^a,b^	22.11 ± 1.76 ^a^	22.06 ± 1.91 ^b^	8.748	0.001	0.097
UKK fitness index	72.95 ± 18.13 ^a,b^	84.72 ± 15.94 ^a^	88.92 ± 19.42 ^b^	12.230	0.001	0.130
VO_2_max (mL·min^−1^·kg^−1^)	19.59 ± 6.14 ^a,b^	23.58 ± 6.05 ^a^	24.51 ± 6.41 ^b^	10.168	0.001	0.111

Note. UKK: Urho Kaleva Kekkonen; BMI: body mass index; VO_2_max: maximal oxygen consumption; m: metre, kg: kilogram; bt: beat; min: minute; mL: millilitre; η^2^: Effect size by eta square; SD: standard deviation; df: degrees of freedom (2, 163 for posttest intervention); ^a,b,c^ superscripts with the same letter show a degree of significant difference between both groups (*p* < 0.05). Pairwise comparisons were performed with Bonferroni’s adjustment.

## Data Availability

Not applicable.

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
