# Peer review of "An Intervention of 12 Weeks of Nordic Walking and Recreational Walking to Improve Cardiorespiratory Capacity and Fitness in Older Adult Women"

_jcm, 2022, doi:10.3390/jcm11102900_

Round 1

Reviewer 1 Report

Title: 12 weeks Nordic walking and Recreational walking to improve cardiorespiratory capacity and fitness in middle-age and older 3 adult women.

After I reviewed this paper, I think this paper needs to add more information for clearly information and research quality improvement.

Abstract: for conclusions: “…both were shown to be valid modalities to improve the functionality and quality of life of people during the aging process.” Only this sentence in the paper mentions about functionally and QOL, I think they are not link to the objective and results of this study. please rewrite it.

  1. Introduction:

Line 98-99: The gap of research and relevant of research should to state in this paragraph prior to objective. This will make readers more understanding of the purpose and importance of this research.

Line 99-103: Please the objective. Suggest to group the primary outcome for mention in the objective. concise

  1. Materials and Methods;

Line 107,108: the term of Mage. Please check the symbol of mean.

Figure1: suggest to revise the flowchart for clearly presentation. If the subject denied to participate in the intervention program, they should not pass through the random allocation process. The same as the other criteria such as age range 50-69y and having somatic condition, these criteria should be screen out before group allocation. There are three groups for comparison in this study, then the flowchart should separate into three arms NW, RW and control. The time to follow-up in flowchart should to specify. 12weeks?

Line 138: body composition variables measurement should include many indicators such as body fat, muscle mass but this study measured body weight and calculated to BMI. Therefore, suggest to change the topic correspond to the parameter of measurement for reducing confusion.

Line 143-144: before the test, Do the subject allow to take the medication for their underlying diseases such as beta-blocker medicine? And How to control the effect of medication on HR?

Line 173-218: Suggest to explain how to reduce the bias of assessors and researchers that might occur in this study for improve your internal validity.

Line 222-223: this is RCT design, should add the approval from Clinical trial registration.

Line 225-243: Statistical analysis. Based on your design, 3 groups comparison and repeated measured over time, the one-way ANOVA and Repeated measured ANOVA used for data analysis separately. this was not appropriate statistics and caused of type I error inflation. Suggest to revise the statistic selection such as Mixed Model ANOVA.

And other issue is how to manage the subject who loss to follow-up? What is the statistic for data analysis with loss to follow-up (6 persons in experimental group), I suggest to use data imputation and ITT.

  1. Results

Table 1: how to reduce the effect of baseline data difference? The results demonstrated that walking HR at baseline were significant difference among 3 groups, these consequence to different at the 12 weeks follow-up.

Table1,2,3: demonstrate the effect size calculation, please explain this indicator and why need to present in this study. Suggest to add in the statistical analysis section.

Should to revise the legend of Table 1, 2 (Note. ….. abcCommon superscripts in the same row indicate a significant difference (p<0.05) between 275 the groups with the same letter.) This sentence is not clear for significant difference of which was pair comparison? Please revise more clearly.

  1. Discussions

Line 417-418: this sentence “Therefore, more research is needed to compare these exercise modalities over longer 417 periods of time and in different segments of the middle-aged and older adult population.” it is not accordance with above sentence. Suggest to rewrite  

Add the limitation of this study in this section. Move Line 430-434 into this section. 

Author Response

Dear Reviewer,

Thank you very much for your help and suggestion. We have tried to improve the manuscript according your comments. Regards.

Title: 12 weeks Nordic walking and Recreational walking to improve cardiorespiratory capacity and fitness in middle-age and older 3 adult women.

After I reviewed this paper, I think this paper needs to add more information for clearly information and research quality improvement.

Abstract: for conclusions: “…both were shown to be valid modalities to improve the functionality and quality of life of people during the aging process.” Only this sentence in the paper mentions about functionally and QOL, I think they are not link to the objective and results of this study. please rewrite it.

Response: Thank you very much for your suggestion. The text has been updated accordingly. Moreover, we have added lines to support our research.

---

  1. Introduction:

Line 98-99: The gap of research and relevant of research should to state in this paragraph prior to objective. This will make readers more understanding of the purpose and importance of this research.

Response: Following the literature research about the topic of the study and a recent systematic review on NW and RW, it has been described in the introduction section that there is evidence, but more research is needed on both physical activity practices.

---

Line 99-103: Please the objective. Suggest to group the primary outcome for mention in the objective. Concise

Response: Thank you for suggestion. The text has been changed accordingly.

---

  1. Materials and Methods;

Line 107,108: the term of Mage. Please check the symbol of mean.

Response: We have changed Mage by μag according to universal statistical symbols.

---

Figure1: suggest to revise the flowchart for clearly presentation. If the subject denied to participate in the intervention program, they should not pass through the random allocation process. The same as the other criteria such as age range 50-69y and having somatic condition, these criteria should be screen out before group allocation. There are three groups for comparison in this study, then the flowchart should separate into three arms NW, RW and control. The time to follow-up in flowchart should to specify. 12weeks?

Response: Thank you very much. We have followed all your suggestion about the flow diagram. However, we have considered to maintain the three groups in the same flowchart because we think that it is clearer to show to readers.

---

Line 138: body composition variables measurement should include many indicators such as body fat, muscle mass but this study measured body weight and calculated to BMI. Therefore, suggest to change the topic correspond to the parameter of measurement for reducing confusion.

Response: Thanks for your suggestion and we have changed “body composition” by “anthropometric measurement” in all manuscript.

---

Line 143-144: before the test, Do the subject allow to take the medication for their underlying diseases such as beta-blocker medicine? And How to control the effect of medication on HR?

Response: Underlying diseases medicine were allowed to take them. Moreover, participants with severe illness and important medicines were not included in the study such as participants with cardiovascular or neuromuscular illness.

---

Line 173-218: Suggest to explain how to reduce the bias of assessors and researchers that might occur in this study for improve your internal validity.

Response: Each specialist controlled a different variable studied (Anthropometric measurement, HR, UKK test measurements and VO2max variables. Thus, bias of assessors and researchers was minimum.

---

Line 222-223: this is RCT design, should add the approval from Clinical trial registration.

Response: Thanks. Of course. We have added the RCT after the inclusion criteria paragraph.

---

Line 225-243: Statistical analysis. Based on your design, 3 groups comparison and repeated measured over time, the one-way ANOVA and Repeated measured ANOVA used for data analysis separately. this was not appropriate statistics and caused of type I error inflation. Suggest to revise the statistic selection such as Mixed Model ANOVA.

Response: Thank you very much for your suggestion. Your proposal was added to the manuscript to elucidate the figures of this study. A Mixed Model ANOVA was performed and attached in the Figure´s explanations by several paragraphs (pag. X to x).

Moreover, to better clarification, we have changed “baseline” by “pretest” and “12 weeks” by “posttest”.

And other issue is how to manage the subject who loss to follow-up? What is the statistic for data analysis with loss to follow-up (6 persons in experimental group), I suggest to use data imputation and ITT.

Thanks. We understand that the Mixed Model ANOVA performed assumes the loss of the 6 persons.

---

  1. Results

Table 1: how to reduce the effect of baseline data difference? The results demonstrated that walking HR at baseline were significant difference among 3 groups, these consequence to different at the 12 weeks follow-up.

Response:

The initial difference between groups may be due to different reasons: i) active move to the laboratory, ii) rest the night before, iii) or even some emotional aspects. etc... It is due to the randomness of the data, in fact they are previously undesired effects that are later controlled after speaking with the study subjects and receive the initial information.

---

Table1,2,3: demonstrate the effect size calculation, please explain this indicator and why need to present in this study. Suggest to add in the statistical analysis section.

Response: We have changed information about cohen d instead of Eta square in the “Statistical analysis” section. In addition, we have added information about the interpretation.

---

Should to revise the legend of Table 1, 2 (Note. ….. abcCommon superscripts in the same row indicate a significant difference (p<0.05) between the groups with the same letter.) This sentence is not clear for significant difference of which was pair comparison? Please revise more clearly.

 Response: The superscripts according to the table have been updated and the description has been modified for better understanding.

 ---

  1. Discussions

Line 417-418: this sentence “Therefore, more research is needed to compare these exercise modalities over longer periods of time and in different segments of the middle-aged and older adult population.” it is not accordance with above sentence. Suggest to rewrite

Response: The paragraph has been rewritten to be consistent with the previous one.

---

Add the limitation of this study in this section. Move Line 430-434 into this section. 

Response: Limitations of the study are now in the discussion section.

Reviewer 2 Report

The article “12 weeks Nordic walking and Recreational walking to improve cardiorespiratory capacity and fitness in middle-age and older adult women” is interesting and the sample size is quite good. It is also a good choice to study only women, as most NW enthusiasts are women.However, there is already previous literature on the effects of NW on CRF.

Abstract:

This part includes for methods part of the abstract.

“Body composition 21 variables were measured by stadiometer and electronic scale.”

Also, why you present your participants in two age group? but you not compare these?

Is it better to say participants were aged 50 to 69 years old

age=57.53+6.85 should be reported by only one decimal

Introduction:

Instead, inactive time authors should use sedentary time or physical inactivity

Authors could mention also good correlation between Urho Kaleva Kekkonen Test (UKK test) and Vo2max test.

Material and methods:

The same comment as in abstract: Why you present your participants in two age group? but you not compare these? How are participants divided in these three groups?

How was this ensured? During this intervention, participants took part in 182
no other physical activity?

Specify what NW poles participants used?

Results:

In table 2 Is it better to analyses change of intervention? or use Ancova because there is baseline difference between groups?

In Table 3 there is clear shift from fitness class but why you did not have analyze it statistically?

A repeated measures ANOVA could also report time and group interaction

Discussion

There is misspelling and what this means? If we compare our results with the findings of xx,??

How was the participant’s Nordic walking technique controlled during study? That could comment on the weakness and strength part of discussion.

Also, your study older adults but you did not compare then for middle-aged women. So, this comparison would be strengthening your study? and discussion.

Author Response

Dear Reviewer,

Thank you very much for your help and suggestion. We have tried to improve the manuscript according your comments. Regards.

Abstract:

This part includes for methods part of the abstract.

“Body composition 21 variables were measured by stadiometer and electronic scale.”

Also, why you present your participants in two age group? but you not compare these?

Is it better to say participants were aged 50 to 69 years old

age=57.53+6.85 should be reported by only one decimal

Response: The age of the study participants has been unified and the number has been corrected to just one decimal.

 ---

Introduction:

Instead, inactive time authors should use sedentary time or physical inactivity

Response: Thank you. The term inactive time has been changed to sedentary time in the manuscript.

---

Authors could mention also good correlation between Urho Kaleva Kekkonen Test (UKK test) and Vo2max test.

Response: Thank you very much. The section was updated.

---

Material and methods:

The same comment as in abstract: Why you present your participants in two age group? but you not compare these? How are participants divided in these three groups?

How was this ensured? During this intervention, participants took part in no other physical activity?

Specify what NW poles participants used?

Response: We present the participants in three groups (control, Nordic walking and Recreational walking) and we add the mean age of each group as extra information. The age group is not studied in any case. It is only extra information. We think that it is interesting to add the age. However, if the reviewer suggest us to delete it we do not have problems to do it.

During the intervention, participants did not take part in other physical activity because they were controlled monthly face to face (Nordic and Recreational group) or by telephone (control group).

According to the poles participants, we have added more information in “Patient and Public Involvement” section.

---

Results:

In table 2 Is it better to analyses change of intervention? or use Ancova because there is baseline difference between groups?

Response: Thank you very much for your suggestion. We have performed a Mixed Model ANOVA to support your suggestion. To better clarification, we have changed “baseline” by “pretest” and “12 weeks” by “posttest”.

---

In Table 3 there is clear shift from fitness class but why you did not have analyze it statistically?

A repeated measures ANOVA could also report time and group interaction

Differences between group and interaction have been performed in Table 1 and Table 2 in the same variable; UKK Fitness Index. However, Table 3 shows subcategories of this test according to scores obtained.

Response: Thanks for your suggestion. However, we have preferred to delete the table 3 to avoid misunderstanding because the information is not relevant.

 ---

Discussion

There is misspelling and what this means? If we compare our results with the findings of xx,?? x

How was the participant’s Nordic walking technique controlled during study? That could comment on the weakness and strength part of discussion.

Response: Thank you. The misspelling has been resolved and extra information has been changed and updated in the Discussion.

---

Also, your study older adults but you did not compare then for middle-aged women. So, this comparison would be strengthening your study? and discussion.

Response: Thanks for your suggestion. We have delete middle-aged women term from the manuscript because the participants ‘age fits better with older adult women.

Reviewer 3 Report

Dear authors,

The area of the research is very new and interesting and valuable. However it needs a few amendments.

Introduction

Please add more about exercise effectiveness in older people in the Introduction. Why is important?

Please say more on Urho Kaleva Kekko-72 nen Test (UKK test). Reliable, valide, sensitive?

‘For this reason, in the present research, we used the Urho Kaleva Kekko-72 nen Test (UKK test).’ This is not correct to be mentioned at this point. Please transfer it in methods.

Make the aim clearer.

Please add new REFs at the introduction and Discussion

i.e

Effects of multicomponent exercise training intervention on hemodynamic and physical function in older residents of long-term care facilities: A multicenter randomized clinical controlled trial. https://doi.org/10.1016/j.jbmt.2021.07.009

i.e. Line 52-56 OR add more in Line 60

Methods

Sample: Please write Mean ± SD

Please write more about sample’s recruitment

Ethical approval?

Discussion

What is the clinical importance?

Any future studies?

Author Response

Dear Reviewer,

Thank you very much for your help and suggestion. We have tried to improve the manuscript according your comments. Regards.

Comments and Suggestions for Authors

Dear authors,

The area of the research is very new and interesting and valuable. However it needs a few amendments.

Introduction

Please add more about exercise effectiveness in older people in the Introduction. Why is important?

Response: The benefits of physical exercise in this population group have been reinforced based on systematic reviews and meta-analysis.

---

Please say more on Urho Kaleva Kekko-72 nen Test (UKK test). Reliable, valide, sensitive?

Response: Thanks. We used the reference 19 to justify the different question about UKK test.

---

‘For this reason, in the present research, we used the Urho Kaleva Kekko-72 nen Test (UKK test).’ This is not correct to be mentioned at this point. Please transfer it in methods.

Response: Thank you very much. The introduction has been changed and updated correctly.

---

Make the aim clearer.

Response: We have changed “Body composition” by “anthropometric measurement variables” for better understanding.

---

Please add new REFs at the introduction and Discussion

i.e

Effects of multicomponent exercise training intervention on hemodynamic and physical function in older residents of long-term care facilities: A multicenter randomized clinical controlled trial. https://doi.org/10.1016/j.jbmt.2021.07.009

i.e. Line 52-56 OR add more in Line 60

Response: Several quotes from recent publications have been updated in Introduction and Discussion

---

Methods

Sample: Please write Mean ± SD

Please write more about sample’s recruitment

Ethical approval?

 Response:

We have changed the mean by an universal statistical symbol as μ +SD

We have added more information about sample´s recruitment and ethical approval.

---

Discussion

What is the clinical importance?

Any future studies?

Response: Thank you very much. The discussion and conclusion section has been changed and updated correctly.

Round 2

Reviewer 1 Report

Thank you for your responses. 

Reviewer 3 Report

It is well improved. Thanks